# Identification of a pocket factor that is critical to Zika virus assembly

Nadia M. DiNunno[1,2], Daniel J. Goetschius [1,2], Anoop Narayanan[1], Sydney A. Majowicz[1], Ibrahim Moustafa[1], Carol M. Bator[3], Susan L. Hafenstein[1,2,3] & Joyce Jose [1,3✉]

Zika virus (ZIKV) is an emerging mosquito borne flavivirus and a major public health concern causing severe disease. Due to the presence of a lipid membrane and structural heterogeneity, attaining an atomic resolution structure is challenging, but important to understand virus assembly and life cycle mechanisms that offer distinct targets for therapeutic intervention. We here use subvolume refinement to achieve a 3.4 Å resolution structure and identify two distinct lipid moieties. The first arises from the inner leaflet and is coordinated by hydrophobic residues of the M and E transmembrane helices that form a binding pocket not previously characterized. The second lipid arises from the outer leaflet coordinate between two E protein helices. Structure-based mutagenesis identifies critical hydrophobic interactions and their effect on the virus life cycle. Results show that lipids play an essential role in the ZIKV assembly pathway revealing a potential target of lipid based antiviral drug development.

[1] Department of Biochemistry and Molecular Biology, The Pennsylvania State University, University Park, PA 16802, USA. [2] Department of Medicine, The Pennsylvania State University College of Medicine, Hershey, PA 17033, USA. [3] The Huck Institutes of the Life Sciences, The Pennsylvania State University, University Park, PA 16802, USA. ✉email: jxj321@psu.edu

ZIKV is a mosquito-transmitted flavivirus first isolated in 1947 from the Zika forest of Uganda[1] that has recently emerged and spread to over 30 countries in the Americas affecting millions of people[2,3]. ZIKV is associated with Guillain–Barré´ syndrome in adults[4,5] and has the potential for significant human-to-human transmission[6,7]. Unlike most other members of the flaviviridae family, ZIKV can cross the placenta during pregnancy and infect the developing fetus to cause congenital malformations, including microcephaly and fetal demise[8,9].

ZIKV has a 500-Å-diameter icosahedral, smooth surface covered by 180 copies each of membrane-anchored envelope (E) (504 amino acids), and membrane (M) (75 amino acids) proteins[10–12]. The M and pr (93 amino acids) proteins are generated from a precursor membrane protein (prM, 168 amino acids) by the host furin protease cleavage at the prM junction. The positive-strand RNA genome of 10,807 bases is packaged by the capsid protein (C) (105 amino acids) inside a host-derived lipid bilayer. Each E and M protein pair form a dimer, with the E–M dimers lying parallel to each other to form a herringbone-like raft comprised of a total of six E and six M, with 30 such rafts covering the virus surface[12,13]. The mature ZIKV consists of E, M, and C structural proteins[14], whereas the spiky immature ZIKV also contains the pre-membrane protein, prM incorporated as 60 trimeric prM-E heterodimers. Flavivirus E protein is involved in receptor binding, attachment, and virus fusion during cell entry[13]. The ectodomain of E protein contains three domains: DI, DII, and DIII, followed by an α-helical stem region that lies on the viral membrane with three stem helices (EH1, E-H2, and E-H3)[10,12] and a C-terminal transmembrane region containing two transmembrane helices (E-T1 and E-T2)[15]. The M protein is mostly unexposed and has three helices (M-H1, M-H2, and M-H3), including two transmembrane helices (M-H2 and M-H3). The ZIKV E protein is glycosylated at Asn 154[12]. The N-linked glycan of E protein is associated with enhanced infectivity via cell surface lectins and has been implicated in neurovirulence[16,17]. The E-DIII is associated with receptor binding[18], and the E-DII contains the fusion loop that interacts with the membrane during low pH-mediated endosomal fusion[15]. The ZIKV pr domain is glycosylated at Asn 69 and cleaved from prM by host protease furin at the trans-Golgi network (TGN) during virus maturation[19]. Interaction between E and M, mediated by pH-sensing histidine residues, is required for preventing premature exposure of the fusion peptide in mature DENV[20]. The pr domain protects the fusion loop on the E protein from nonproductive interactions within the cell[21,22]. The removal of the pr peptide during maturation exposes the fusion loop and primes the virus for subsequent conformational changes necessary for membrane fusion.

Flaviviruses assemble initially as immature, spiky, non-infectious particles with a diameter of approximately 600 Å. Upon budding into the endoplasmic reticulum (ER), they are transported through the secretory pathway[19,23,24] where prM and E proteins undergo pH-induced changes. With these conformational changes, the spiky immature virus containing 60 trimers[10,19] becomes a smooth mature particle containing 90 dimeric M-E heterodimers[12,13]. Additionally, the immature ZIKV cryo-EM structure at 9-Å resolution showed a partially ordered capsid protein shell close to the transmembrane domains of the E and M proteins that is less prominent in other immature flaviviruses and absent in all mature flaviviruses, suggesting a rearrangement of the capsid shell during maturation[19]. However, most flavivirus capsid cores appear to lack a discernable organization[24] although a recent cryo-EM structure of immature ZIKV has shown the orientation of capsid protein and its interactions

with the transmembrane regions of M and E[25]. Flaviviruses undergo proteolytic processing by the cellular protease furin in the TGN, and the pr domain is cleaved from the prM protein, which remains associated until the virion is released to the extracellular milieu[22]. Although the immature and mature viruses have different surface protein organizations, their envelope protein transmembrane (E-TM) regions remain in similar positions during maturation[10]. The mature particles that are released from the cells by exocytosis release the pr peptide at neutral pH, enter the cell using receptor-mediated endocytosis, and fuse with the endosomal membrane at low pH, releasing the RNA genome[26].

There are significant gaps of knowledge in the assembly and maturation processes of flaviviruses in general and ZIKV in particular. Although we know that large structural rearrangements are required to make the necessary conformational changes, there is little understanding of how this process occurs. For example, the molecular mechanisms leading to the genomic RNA packaging by C protein and encapsidation of nucleocapsid by the viral envelope at the ER budding site are incompletely understood. Furthermore, we know little about the viral and host factors involved in the structural transition from immature to mature. Previously, structural studies to address these unknowns have focused on traditional processing methods, including increased particle number and higher magnification to overcome global and local heterogeneity. Here, we used a cryo EM reconstruction approach in which subvolumes of the virus were designated, extracted, and refined, such as in block-based and localized reconstruction approaches[27–29]. Our structure improved in resolution to 3.4 Å, and we discovered two distinct lipid moieties coordinated by M and E helices that have not previously been described. We characterized a hydrophobic binding pocket and a binding cleft not recognized before. The resulting atomic structure was used to guide a mutational analysis to understand the mechanism of lipid incorporation and its importance to ZIKV assembly. This novel discovery opens a new avenue for drug development with the potential to combat ZIKV specifically and flavivirus in general.

## Results

**The icosahedrally averaged ZIKV map attained 4.0-Å resolution.** ZIKV (MR-766, Uganda strain) was grown in Vero-E6 cells, purified, and vitrified for the data collection on the Pennsylvania State University Titan Krios. The raw micrographs showed heterogeneity of virus particles, as has been reported previously[10,12] (Fig. 1a). The representative 2D micrographs and classes illustrated well the patches of uncleaved prM that are consistently found throughout the data (Supplementary Fig. 1). From the micrographs, 33,653 particles were selected for the reconstruction. The subpopulation of the mature virus was further classified during the single-particle reconstruction that was done while imposing icosahedral symmetry averaging. The 4.0 Å resolution icosahedrally averaged map (Fig. 1b) showed similar features as the previously solved structures of mature ZIKV (PDB ID: 5IZ7 and 6CO8)[10,12]. E-glycoprotein subunits and buried M protein alpha helices were resolved embedded in the lipid bilayer (Fig. 1c). The nucleocapsid was not resolved presumably due to the lack of icosahedral symmetry in the mature virus. The alpha-helical domains of the E and M proteins were clearly visible, although the glycosylation of the E protein was not resolved.

**Subvolume classification and reconstruction improved map quality and resolution.** Asymmetric patches of uncleaved prM introduced local heterogeneity that could not be overcome using traditional cryo-EM reconstruction approaches that average the

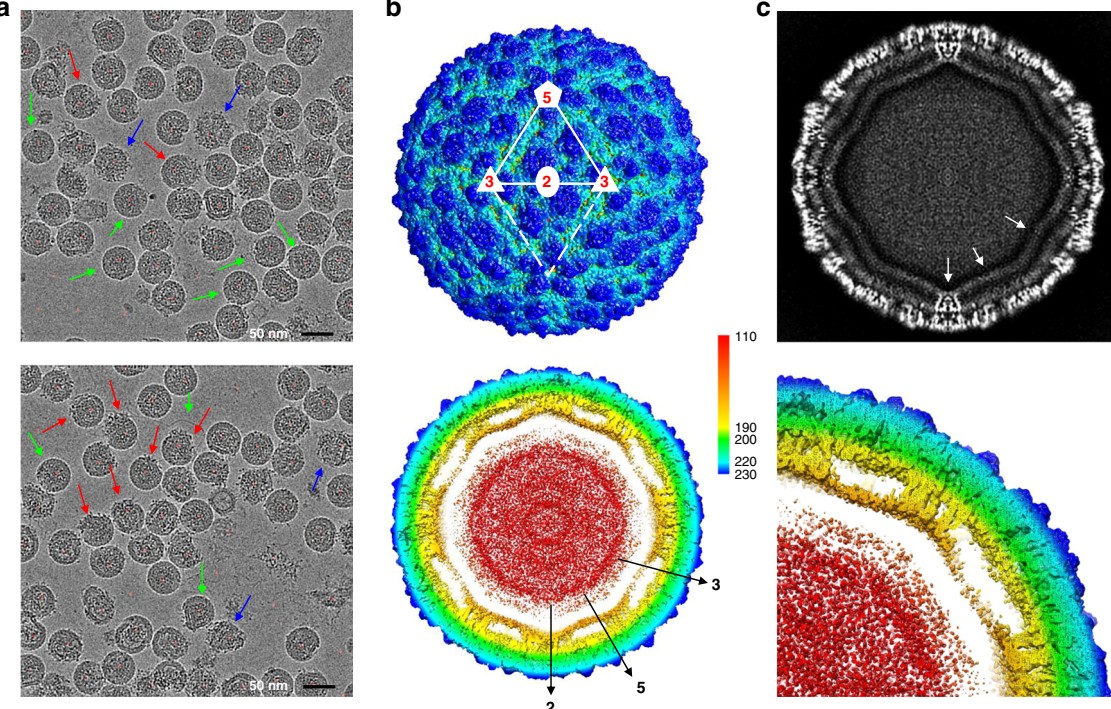

**Fig. 1 Cryo-EM single-particle reconstruction of the mature Zika virus. a** Representative micrographs from 2128 total micrographs in the dataset of vitrified MR-766 ZIKV purified from Vero-E6 cells. Micrographs contained particles with large regions of immature, uncleaved prM (blue arrow), mosaic immature (red arrows), and mostly mature particles (green arrows). **b** Surface rendered density map of the icosahedrally averaged reconstruction with two asymmetric units (white solid and dashed lines) and symmetry axes (red font) labeled to highlight the raft subunit comprised of three EM dimer pairs (top). The slabbed map (bottom) is radially colored according to the key ($r$ = 110–230 Å) to emphasize the layering of the mature virion, including glycoprotein (blue–green), lipid bilayer (yellow), and capsid protein, and genome RNA (red). **c**, top: A central section illustrates the quality of the density in the 4.0 Å 60-fold symmetric reconstruction. **c**, bottom: The zoom quarter map section shows the conserved transmembrane E–M helices within the lipid bilayer (yellow).

subunits together over the entire virus. Computational approaches such as block-based and localized reconstruction isolate smaller subvolumes for reconstruction can overcome this type of heterogeneity and improve the resolution (Supplementary Fig. 1). To attain high-resolution and elucidate crucial protein interactions, we used a subvolume reconstruction approach[27,30]. Density corresponding to two asymmetric units was designated as a subvolume, which specifically consisted of dimers of three M and E-protein pairs straddling the twofold symmetry axis. For each virus, 30 twofold subparticles underwent 3D classification and local refinement to reduce the effect of asymmetric heterogeneity. The final map reached 3.4 Å after postprocessing (Supplementary Figs. 2 and 3). The improvement in resolution allowed unambiguous identification of the main chain and most side-chain positions throughout the structure. The atomic structure of ZIKV (PDB ID 6CO8) was used to initiate model building followed by a refinement, in which side chains and residues were moved as needed to fit density appropriately with final verification of the model in Molprobity (Supplementary Table 1)[31–33]. The C-alpha atoms superimposed with the previous structure solution (PDB ID 6CO8) with a root-mean-square deviation of 0.94 Å[2]. The transmembrane helices of M and E proteins were clearly differentiated along with their respective side chains. Surprisingly, an unfilled density was observed arising from the inner leaflet and stretching down between the M and E helices positioned such that there was one unfilled density per each E–M protein heterodimer (Fig. 2). A second density was found between E transmembrane helices arising from the outer leaflet (Fig. 2).

**The densities unfilled by virus protein are lipids**. Both unfilled densities were equal in magnitude to the virus structural protein density and had no side chains. The density that arose from the inner leaflet originated with a crook and extended along the length of M and E helices measuring ~28 × 3 Å (Supplementary Fig. 4). This density was coordinated by transmembrane helix residues M F42-A48, M T57-I70, E W462-L477, and E L489-S499. The lipid density was fitted with hydrocarbon chains of varying lengths to test the possibility that a fatty acid group was incorporated. The density accommodated a 20 Å tail of ~16 hydrocarbons in length with an 8-Å kinked-head density that could fit another 8 hydrocarbons.

The Y-shaped density that arose from the outer leaflet extended inward ~10 Å before branching into 5 and 8 Å arms, all of which were bound between two E helices stretching between S405-L424. The specific residues interacting with this lipid included E H446, E F449, E F453. Fitting experiments showed that hydrocarbon chains of about 10, 5, and 7 could be incorporated into the arms of the Y, which may best accommodate a phospholipid. In both cases, the transmembrane helices of the virus proteins provided a hydrophobic environment, thus creating hydrophobic pockets into which the lipids fit.

A comparison with previously published ZIKV maps revealed that regardless of the virus origin, the Y-shaped lipid was incorporated at the outer leaflet of the lipid bilayer; however, the only virus grown in mammalian cells had the crooked lipid density arising from the inner leaflet (H/FP/2013 strain) (PDB ID 6CO8)[12] (Fig. 2). Due to the absence of the lipid in virus propagated in mosquito cells, we compared the pocket structures between our map and that of PDB ID 5IZ7[10]. We found that the pockets are nearly identical with the exception of the side-chain density corresponding to Trp 474, which swings into the lipid-filled pocket, but away from the empty pocket (Fig. 2). The

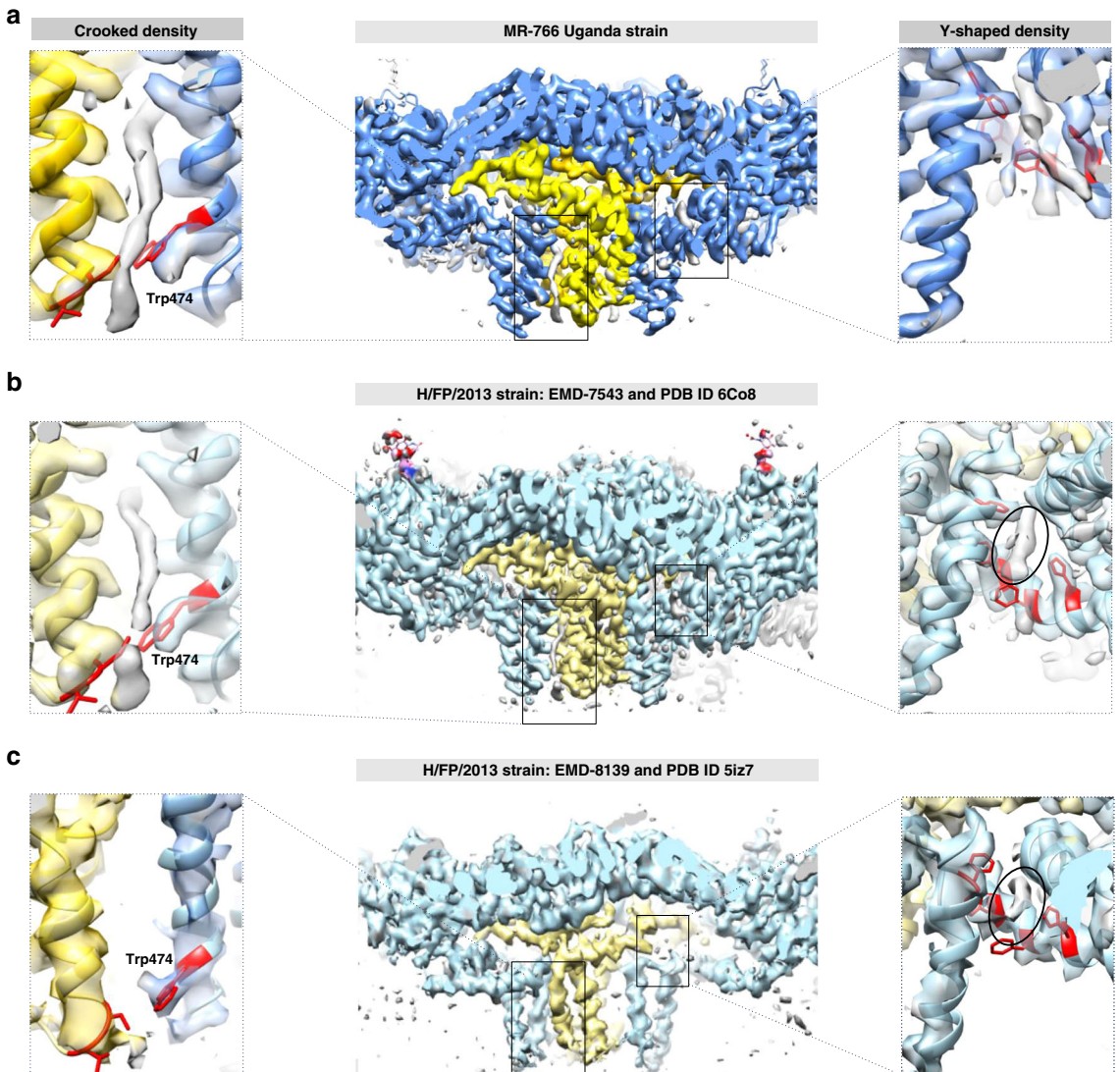

**Fig. 2 Comparison lipid-binding pockets of ZIKV structures.** To compare our structure (**a**) to EMDB 6CO8 (**b**) and 5IZ7 (**c**), a zoomed view of the cross sections of each map was surface rendered to illustrate two M proteins (yellow) and two E glycoproteins (shades of blue). Both maps presented in panels **a** and **b** were reconstructed from particles propagated in mammalian cells, whereas the third map (**c**) was derived from mosquito cells. **a** The crooked lipid density (gray, left zoom panel) shown stemming from the inner leaflet of the lipid bilayer is bound into a hydrophobic pocket formed by the M and E transmembrane helices (yellow and blue, respectively). **a**, right zoom panel: The Y-shaped lipid density (gray and black circles) is located between the amphipathic helices of each E glycoprotein. These lipid densities are represented twice in each heterodimer between M and E proteins. The density maps of the MR-766 strain of ZIKV (**a**) and the [6CO8] French Polynesian strain (**b**), both propagated in mammalian cells, have the crooked density bound in the pocket between the M and E-glycoprotein alpha helices. However, the lipid density is absent from the [5IZ7] French Polynesian map (**c**), which was propagated in mosquito cells. The Y-shaped lipid density was identified bound into the pocket in all structures. **a–c** Residues interacting with the densities are identified in red and include S58, T57, W474, F449, F453, F446, F487, according to MR-766 numbering. The critically important Trp 474 is labeled (left, zoom panel).

resolution varied somewhat among the different structures we were comparing. To test if the lipid density was dependent on map resolution, we rendered our map to 3.7 Å resolution and found that the lipid density was still obvious and bound between the transmembrane alpha helices at the interaction site. These findings suggest that the incorporation of the crooked shaped lipid occurs in mammalian cells and not insect cells.

**The presence of the lipid is critical to function.** To test the significance and understand the mechanism of ZIKV lipid incorporation, we used the cryo-EM structure to guide a mutagenesis study (Fig. 3). For the crooked lipid, the amino acid side

chains subjected to mutagenesis were selected from M (Thr 57 and Ser 58) and E proteins (Trp 474). For the Y-shaped lipid, mutagenesis-targeted residues E H446, E F449, E F453. Initially, the following alanine substitutions were made in M protein (M T57A, and M S58A) and E protein (E H446A, E F449A, E F453A, E W474A, and E F497A) using a full-length ZIKV cDNA clone. Plaque assays were performed to assess the effect of the mutations on the virus life cycle. Alanine substitutions of M residues T57A and S58A did not affect virus production, while alanine substitution of E-protein residues (F449A, F453A, H446A, W474A, and F497A) were lethal for ZIKV. The amino acid Trp 474, which is located near the inner leaflet of the lipid bilayer and presumably

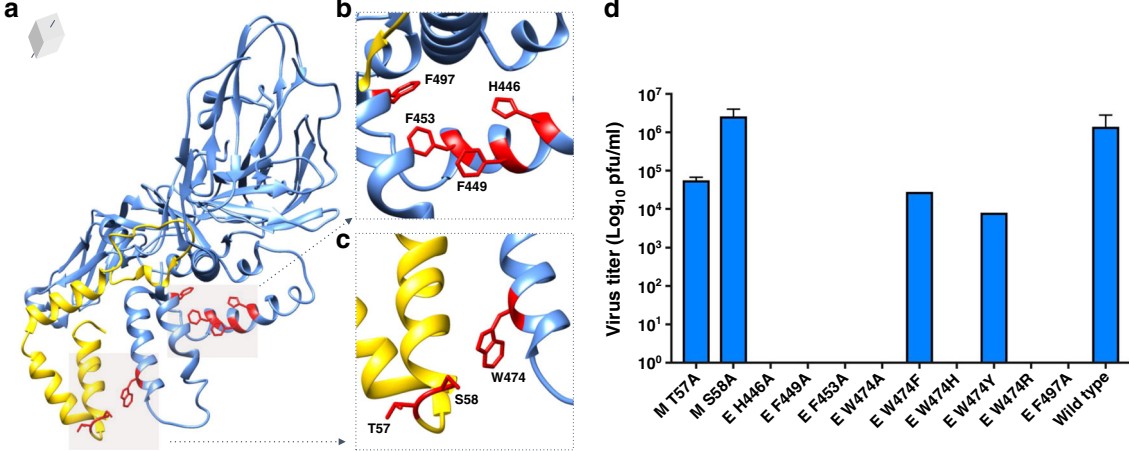

**Fig. 3 Structure-based mutagenesis demonstrates lipid interactions are essential for virus release. a–c** The MR-766 M and E proteins (yellow and blue ribbon, respectively) illustrate the location of residues interacting with the lipid densities that were chosen for mutational analysis (red). **d** Plaque assays of the mutant viruses compared to wild-type MR-766 virus showed that all but four mutations were lethal. Data shown are from three technical replicates of one biological experiment, and the data are expressed as the mean with standard error (SEM).

contacting the headgroup of the lipid density, was selected for further mutagenesis. To test which side-chain identities would be tolerated near the lipid headgroup, the following mutations were generated: E W474H, E W474Y, E W474R, E W474F (Fig. 3). Single-amino-acid substitutions of E W474H and E W474R produced a lethal phenotype, whereas E W474F and E W474Y did not significantly affect virus production and gave rise to medium plaques. Thus, the mechanism of lipid incorporation depends on maintaining the hydrophobicity of the newly identified inner leaflet pocket. Additionally, the presence of both the crooked lipid in the pocket and the Y-shaped lipid in the hydrophobic cleft were critical for the ZIKV life cycle.

## Discussion

In the mature virus, the lipid bilayer is sandwiched between the outer M and E proteins and the inner disordered capsid protein and RNA genome. Although the outer proteins display obvious icosahedral symmetry, the lipids remain somewhat fluid, differing in thickness and angularity relative to the protein layers. Hence, each raft (consisting of three M and E dimer pairs) may not maintain strict icosahedral symmetry relative to its neighbor unless maturation has been achieved globally on each particle. Often ZIKV is propagated in Vero-furin cells that overexpress host protease furin to obtain a homogenous population of mature virus[11,12]. In this study, we have used the mammalian cell line, Vero-E6, that do not overexpress furin, resulting in particles that displayed patches of immature spikes throughout (Supplementary Fig. 1). We demonstrated that a local reconstruction approach could compensate for this type of asymmetry and improve the icosahedrally averaged map. This improvement in resolution and map quality suggests that the extraction and refinement of sub-volumes overcome imperfection in the icosahedral symmetry of ZIKV from mammalian cells. Importantly the map improvement is what allowed the identification of distinctly bound lipids, the hydrophobic pocket, and the Y-shaped lipid-binding cleft. A similar cryo-EM approach for the Sindbis virus revealed a hydrophobic pocket with bound pocket factor[34]. Previous crystallographic studies of the DENV2 E protein identified a hydrophobic pocket formed by E proteins (ED-II and E-DI pocket)[35]. We propose a structural role for the pocket and suggest it stabilizes mature E. The phenomenon of virus evolving to use a hydrophobic pocket to bind a lipid moiety to enhance stability has also been seen in other viruses[36]. A recent study describing

the cryo-EM structure of mature Spondweni virus (SPOV) has identified the presence of a density strongly resembling the headgroup of a phosphatidylethanolamine lipid (PE), with partially resolved fatty acid tails, in a pocket formed near the amphipathic stem helices of E[37]. Hydrophobic residues surrounding the lipid density was described as important for the virus life cycle with Phe 454, Leu 499, and a histidine His 447 (SPOV residue numbers) packing against the lipid in close vicinity to the phosphate group of PE. Consistent with our observation, the high-resolution SPOV structure was propagated in mosquito cells and did not incorporate the crooked lipid near the TM helices[37].

Regardless of how the virus was propagated, the Y-shaped lipid was bound at the outer leaflet, and amino acid substitutions that would affect lipid incorporation were lethal. Thus, the bound lipid seems essential, and the location suggests a role in E-protein stabilization or maturation. Only mammalian cell-cultured ZIKV incorporated the crooked lipid into the hydrophobic pocket arising from the inner leaflet. For a comparison with structures of other flaviviruses, we examined the 3.5-Å resolution dengue virus map (PDB ID 3J27)[38] and 3.9-Å resolution map (PDB ID 5O6A) of tick-borne encephalitis virus[39], but in neither case could the presence or absence of bound lipid be determined. The presence of this lipid and the critical nature of the residues suggest that the occupation of the pocket is necessary to function in mammalian, but not insect cells. Likely, the lipid that fills the pocket is only available in mammalian cells and not produced in insect cells. There are decidedly different pressures on the virus in the mosquito environment compared to mammalian, since the virus must merely survive the mosquito, but thrive in mammals. In the human host, ZIKV must propagate to high titer producing a viremia to successfully transmit progeny. The virus has been detected in semen, vaginal fluids, saliva, and urine, suggesting there is an additional requirement for enhanced ZIKV stability to survive in the human host until successful transmission.

**The pocket may be a conserved feature of flavivirus**. Comparison of the M and E-protein sequences from other flaviviruses encompassing the lipid-binding pockets showed the conservation of hydrophobic residues interacting with lipid density (Supplementary Fig. 5). Multiple sequence alignment indicates that the E-protein residues are highly conserved, whereas the M protein residues are not as well conserved. This finding agrees with our

mutagenesis result that also showed E protein more important to function and suggests E governs a fundamental mechanism for lipid binding for all flaviviruses.

At the bottom of the pocket is a key Trp 474 residue that must retain hydrophobic character for virus assembly. For this Trp 474 that is conserved across all flaviviruses, the side chain is pointing away from the headgroup of the lipid moiety and from the pocket formed by the four TM helices of M and E proteins. Trp, which has a strong preference to be at the ends of transmembrane alpha helices near the membrane–water interfacial regions of membrane proteins, has been attributed to have a significant anchoring role besides being part of the TM helix[40,41]. Trp at the ends of TM helices can influence the mismatch between the hydrophobic length of the peptide and the bilayer thickness, which can alter the helix tilt angle[42,43]. Thus, the interfacial aromatic residue Trp situated at the lipid–water interface, is essential for the insertion, stability, and dynamics of transmembrane helices[44]. Possibly, E W474 has such a role in anchoring the E-protein TM helix on the ER membrane. A conserved Trp W409 residue from TM helix of the alphavirus envelope protein E1 has been shown to be part of a pocket that coordinates an unknown pocket factor near the viral membrane in Sindbis virus[34]. The pocket factor molecule has been shown to stabilize the pocket by forming several hydrophobic interactions with surrounding proteins, including the amino acids E1 W409[34]. Therefore, the Trp 474 side chain in ZIKV is ideally positioned to form interactions with lipid pocket factor and other proteins close to the viral membrane and presumably participating in virus assembly. An essential effect on assembly has been seen by cholesterol binding into a pocket in the transmembrane region of influenza virus HA protein[45]. Possibly a hydrophobic interaction between the E and C proteins is required for ZIKV assembly. It has been challenging to find evidence that there is a physical connection between membrane-anchored structural proteins and the C protein or RNA. However, previous cryo-EM research suggested that such a link might be transient. Interactions may form during particle formation that are difficult or impossible to find upon maturation[9].

Recent publications have identified lipid targeting agents as effective ZIKV antivirals, particularly statin derivatives[46]. Our work identifying individual lipid moieties with the potential to regulate assembly and maturation provides insight into the mechanism of statins as antivirals. More importantly, due to the conservation, this antiviral action may extend to other flaviviruses. Importantly, we have identified key residues and functional virus features that were unknown before. This finding opens a new avenue to drug discovery and antiviral therapy to control this family of human pathogens.

## Methods

**Cell culture and purification of ZIKV.** Vero-E6 (African green monkey kidney) and HEK293-T (human embryonic kidney) cells were grown and maintained at 37 °C and 5% $CO_2$ in Dulbecco's modified Eagle's medium (DMEM, Gibco, #12800-082) supplemented with 10% fetal bovine serum (FBS, Seradigm #1500-500) and nonessential amino acids (Gibco, #11140-050). ZIKV MR-766 was obtained from BEI resources NIAID, NIH, as part of the WRCEVA program (ZIKV, MR-766, Uganda strain NR-50065). ZIKV stocks were prepared by growing the virus on Vero-E6 monolayers. After 5 days of incubation, the virus from cell-culture supernatants was collected and stored at −80 °C, and virus titer determined by plaque assay as described below. For virus preparation, a monolayer of Vero-E6 cells grown in DMEM medium supplemented with 5% FBS was infected with the virus at MOI of 0.1. After 12 h, the media over cells were replaced with DMEM with 2% FBS and incubated for 5 days at 37 °C. ZIKV was purified from culture supernatant, as described previously[47]. Briefly, culture supernatant was precipitated with 8% PEG8000 for 4 h at 4 °C, and the precipitate containing the virus was collected after centrifugation at 10,000 × $g$ for 50 min on a sucrose cushion. The precipitate was resuspended in TNE buffer (10 mM Tris HCl pH 8, 140 mM NaCl, and 1 mM EDTA) and subjected to density-gradient centrifugation using a 10–35% tartrate step gradient at 110,000 × $g$ for 2 h at 4 °C. The virus fraction was collected from the gradient, concentrated, and buffer exchanged into TNE using Amicon centrifugal filters (MilliporeSigma, # ACS510024. The purity of the final virus preparation was analyzed by SDS gel electrophoresis.

**Production of mutant ZIKV.** For genetic manipulation of the ZIKV, a cDNA clone of ZIKV strain MR-766 under the control of a CMV promoter (a gift from Dr. Mathew Evans) was used[48]. Mutations of the selected residues were introduced in the cDNA clone by site-directed mutagenesis using Phusion DNA polymerase (NEB, #E0553S) and oligonucleotide primers (IDT) as in Supplementary Table 2. The PCR products were digested with DpnI (NEB, #R0176S) and transformed into *Escherichia coli* K12MC1061. Plasmids were prepared from overnight cultures of the colonies grown in Luria Bertani medium using Qiagen miniprep kit (Qiagen, #27104), and the sequences were confirmed by sanger sequencing at the Sequencing Core facility at The Pennsylvania State University. HEK293-T cells in Opti-MEM media (Gibco# 31985-070) were transfected with wild type and mutated cDNAs using PEI 25 K (Polysciences, # 23966-1). At 12 h post transfection, the culture media were replaced with DMEM supplemented with 10% FBS and incubated at 37 °C. Culture supernatants containing ZIKV were collected at 5 days post transfection and stored at −80 °C.

**Plaque assay and fluorescent focus assay.** Approximately $3 \times 10^5$ Vero-E6 cells were seeded in six-well dishes and grown to form a confluent monolayer. ZIKV stocks were serially diluted in PBS supplemented with 1% FBS and 1 mM each of $CaCl_2$ and $MgCl_2$, and 250 μL were added to Vero-E6 cells in six-well plates and incubated at room temperature for 1 h with rocking. The monolayers were overlaid with 3 ml of 1% agarose in DMEM and incubated at 37 °C with 5% $CO_2$. After 5 days, the virus titers were determined by staining the plates with neutral red (MilliporeSigma, #N2889) and counting the number of plaques. ZIKV stocks were serially diluted in PBS supplemented with 1% FBS and 1 mM each of $CaCl_2$ and $MgCl_2$, and 10 μL were added to confluent Vero-E6 cells in 96-well plates and incubated at room temperature for 1 h with rocking. The cells were incubated at 37 °C with 5% $CO_2$ for 36 h, and the fluorescent foci were counted by visualizing under a fluorescent microscope.

**Negative staining and cryo-EM sample preparation.** A purified preparation of virus at a volume of 3 μL was applied to freshly glow discharged continuous carbon grids, washed, stained with 2% phosphotungstic acid for 15 s, blotted, and air-dried. Negatively stained ZIKV were screened to assess sufficient quality and concentration for vitrification and high-magnification data collection. For vitrification of purified virus for high-magnification imaging, a QUANTIFOIL R2/1 grid (QUANTIFOIL, Germany) with a thin layer of carbon support was glow discharged using the PELCO easiGlow Glow Discharge System for TEM Grids (Ted Pella, Redding CA). Using the vitrification robot (Vitrobot; Thermo Fisher), sample aliquots of 3.0 μL were applied to grids and incubated for 30 s before plunging into liquid ethane. Grids were stored in liquid nitrogen until screening on the Pennsylvania State University Titan Krios (Thermo Fisher) to determine optimal ice thickness and sample concentration for data collection.

**Cryo-EM data collection.** Images of vitrified ZIKV were recorded on the Krios using EPU Software at a magnification of ×59,000 at 300 kV with spherical aberration correction providing a pixel size of 1.11 Å/pixel. Image stacks (*.mrcs) of 48 image fractions were collected with a total electron dose of 0.87 e-/Å²/frame for a total of 41.7 e-/Å² using a Falcon III Direct Electron Detector with a defocus range of −1.0 to −3.0 μm. The diameter of the beam was 1.099 μm, and the exposure time was 66.096 s (Supplementary Table 1).

**Image processing.** Fraction alignment and dose weighting were performed on 3739 micrographs using GCTF and MotionCor2 in the Relion software packages. Particles with poor ice quality and significant astigmatism were excluded from further processing. CryoSPARC[49,50] was used for automated template picking using a preliminary 2D classification template. The initial 3D model was created in RELION using I1 symmetry. Subsequent rounds of 2D classification and 3D classification were performed in RELION on 33,653 particles. Local CTF refinement and polishing were performed in RELION 3.1. Subparticles were designated to contain the density of a raft subvolume (three M and E dimer pairs) and extracted from particle images using the ISECC software package (available for download at www.hafensteinlab.com). This package is a custom implementation of the localized reconstruction approach[27], which optimizes the selection of subvolumes from a larger icosahedral map and then correlates their location onto the original 2D projections. After extracting subvolumes, an initial 3D classification was performed to remove subvolumes that were too divergent from expected orientation, such as immature patches assigned incorrect Euler angles. A total of 9181 subparticles underwent 3D classification in RELION. The best 3D class was refined with limited angular searches resulting in a final resolution of ~3.7 Å. The map reached 3.4 Å after postprocessing. The structure was further refined after initiating a build with the 3.1 Å structure (PDB ID 6CO8) using Phenix (version 1.17)[12]. The refined model was visually inspected in Coot and adjusted to improve rotamer fitting, residue variation, and overall fit. Finally, the structure was validated by MolProbity[31,32,51] (Supplementary Table 1). Root-mean-square deviation

(RMSD) values for bonds and angles were reported at 0.004 and 0.644 accordingly. True Ramachandran outliers were reported at 0.52%. Figures and the difference map were generated using Chimera.

**Reporting summary**. Further information on research design is available in the Nature Research Reporting Summary linked to this article.

## Data availability

Data that support the findings of this study can be requested from the corresponding author upon a suitable request. The cryo-EM maps of the refined subvolume map and the icosahedral map are deposited in the EM data bank under accession numbers EMD-22526 and EMD-22527. The ZIKV model is deposited in the PDB under ID 7JYI.

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

## Acknowledgements

The authors thank the staff of The Pennsylvania State Huck Institutes of the Life Sciences Cryo EM Facility. We thank Matthew Evans for the generous gift of the ZIKV cDNA clone. Funding was provided by The Pennsylvania Department of Health CURE funds. The research reported in this publication was supported by the Office of the Director, National Institutes of Health, under Award Numbers S10RR031780 (S.L.H.), as well as NIH grants R01AI107121 (S.L.H.). The content is solely the responsibility of the authors and does not necessarily represent the official views of the National Institutes of Health. In addition, support for this work was provided by The Pennsylvania State University Start Up Funds (J.J.).

## Author contributions

N.M.D., A.N., S.L.H., and J.J. conceived the study. A.N. and S.A.M. conducted the cell culture, biochemistry, mutagenesis, and virology. C.M.B. prepared the sample for cryo EM data collection and collected the cryo-EM data. D.J.G. designed and developed the custom software. N.M.D. solved the structures, interpreted the maps, and built the models. N.M.D., A.N., I.M., S.L.H., and J.J. interpreted the data. N.M.D., S.L.H., and J.J. wrote the paper.

## Competing interests

The authors declare no competing interests.
