## [Peer Review File · Nature Communications]

Reviewers' Comments:

Reviewer #1:

Remarks to the Author:

In this manuscript, matured Zika virus from Vero-E6 was reconstructed to a 3.4 Å map. Two distinct lipid moieties were identified in a pocket formed by M and E transmembrane helices near the inner leaflet and between two helices of E protein near the outer leaflet, respectively. The density features of Y-shaped lipid can be found but ignored in a previously published map. Using Structure-based mutagenesis, the author showed both lipid moieties play important role in the virus life cycle. The finding is new and interesting.

Major comments

1. The mutagenesis study of E W474 indicates a hydrophobicity of the newly identified inner leaflet pocket. It seems W474 is part of the transmembrane helix, which explains why it requires hydrophobicity. Thus, the hydrophobicity is not just for the crooked lipid. In addition, only virus grown in mammalian cells had the crooked lipid density arising from the inner leaflet also indicates that crooked lipid is not that important since the virus from other cells do not need it. Thus, "Results show that lipids play an essential role in the ZIKV assembly pathway revealing a new target of lipid based antiviral drug development" is not convincing. However, the mutagenesis study of the function of Y-shaped lipid seems more convince to me. Perhaps, Y-shaped lipid plays an essential role in the ZIKV assembly pathway.

2. The authors used a single particle reconstruction approach combining with subvolume refinement to overcome resolution limitations of matured Zika virus and improved the resolution of the map from 4.0 to 3.4 Å. The authors think that the resolution of matured Zika virus is limited by heterogeneity as seen in the reference #10 and #12 in the manuscript. However, in reference 12, the matured Zika virus has been refined to 3.1 Å without subvolume refinement. In addition, a most recent preprint manuscript titled by "A high resolution view of an adolescent flavivirus" on bioRxiv reported matured SPONV and DENV at 2.6 Å and 3.1 Å, respectively. It seems there is no such a limited resolution, since the matured Flaviviruses are rigid enough to achieve high resolution. Thus, it is not suitable to use "resolution limitations" here, probably "improves the resolution" is more suitable. Incomplete cleavage of PrM resulted in heterogeneity of viruses by leaving some of the PrME immature spikes on the mature viruses. However, such particle can be partially avoided during particle picking or later excluded by a careful 3D classification procedure. In this manuscript, ~34,000 particles are too many to achieve a 4.0 Å 70nm icosahedral virus map, which indicates a lot more virus particles can be excluded from refinement. However, instead, the authors used 3D classification of subvolume to get rid of bad particles. As mentioned in the manuscript, only ~9181 (1%, 9181/30/34000=1%) subvolume was selected for refinement. It seems to me that it is not an efficient way to calculate matured Flaviviruses. During the refinement of the whole virus, the half-matured half-immatured particle is very possibly assigned by a wrong Euler angle. In this case, the local refinement of the subvolume of the matured part of the virus cannot help to find the right Euler angle. Thus, the reason for resolution improvement is different from fixing the flexibility problem of the virus. Therefore, the discussion part "each raft (consisting of three EM dimer pairs) may not maintain strict icosahedral symmetry relative to its neighbor" is possibly incorrect.

Minor comments

1. The Y-shaped lipid was found in SPONV and DENV (bioRxiv preprint doi: <https://doi.org/10.1101/2020.06.07.138669>). The preprint was posted on June 7, 2020. They do not mention the other molecule. It may worth to compare.

2. "Previously, structural studies to address these unknowns have focused on traditional processing methods, including increased particle number and higher magnification to overcome global and local heterogeneity. Here, we used a cryo EM reconstruction approach in which subvolumes of the virus were extracted and refined to 3.4 Å resolution."

Although, the package used here is custom implementation of the localized reconstruction approach, the idea here separating the virus into blocks and locally refining the center and orientation parameters of each block separately using the densities of target protein overlapping with the densities of other proteins (without density subtraction) to overcome the flexibility of the icosahedral viruses is closer to that of "block-based" reconstruction method. In the package of the "block-based" reconstruction method, it provided a tool to put the "subvolume" together to form an intact virus. Both "block-based" and "localized method" are well established to deal with flexibility and heterogeneity. It is better to mention and cite both "block-based" and "localized method" methods in the introduction.

Xinzheng Zhang

Reviewer #2:

Remarks to the Author:

The manuscript by DiNunno et.al., identifies the presence of lipid moieties that insert and possibly interact with the transmembrane helices (TM) of E and M proteins of Zika virus. The authors have followed up their structural results with point mutations in the TM regions of E and M proteins to identify critical residues that are likely to be important for interaction with the lipid moieties by analyzing the effect of the mutations on viral infectivity through plaque assays. The manuscript is overall written clearly and concisely. Few comments and questions for the authors are below:

1. The authors have used sub-volume classification and reconstruction to improve the resolution of the map beyond what they achieved by icosahedral averaging alone. This helped them assign side-chain densities with more confidence in the structure. However, the explanation of the sub-volume classification procedure in writing and in extended data figure 1 is a bit confusing. In the text, the explanation implies that sub-volumes of 2 asymmetric units related by 2-fold were selected as a unit, mapped back to their locations in original 2D projections and then were used for subsequent 3D classification of the sub-particles individually. But extended figure 1 shows the whole virus being split into 6 classes. Extended data figure 2 is bit more clearer and seems to align with the local sub-volume classification routine. As this procedure is an important point in the manuscript, it needs to be more clearly explained.

2. Considering that there is a higher resolution reconstruction of Zika virus at 3.1 angstroms available; the authors have used local reconstruction to push their map resolution to 3.4 angstroms to identify amino acid orientations correctly. It would be interesting if the authors compared the local resolution at the TM regions between the maps. Does their approach improve local resolution at the TM region for better model building? Extended data figure 2, shows local resolution variation in top view but not in side-view which may also help to see how the local resolution fares in the TM regions.

3. The authors have compared their ZIKV reconstruction with other ZIKV reconstructions, one at 3.1 angstroms and the other at 3.7 angstroms resolution. They observe that the lipid density arising from the inner leaflet is only seen in viruses grown in mammalian cells and not seen in viruses grown from mosquito cells. Have the authors looked at high resolution structures of other flaviviruses apart from Zika to see if these lipid moieties are seen in other flaviviruses also? The authors have done sequence alignments to show the conservation of important residues in the TM regions of M and E protein among flaviviruses, it would be more informative to see if other maps show similar lipid densities or

not.

4. Overall, figures need to be annotated more clearly and legibly, so that they are more self-explanatory. Example fig.1 – none of the symmetry axes are labeled nor the different colored layers of the cross-section given a scale bar or notations except in the figure legend. In figure 2, it would be nice to label the central panels that are from different EMDB depositions, so that it is clear what the figure is portraying. Similar issues with extended data figures also which lack clear notations. It will also be helpful if the same coloring scheme for E and M proteins are followed across figures.

Reviewer #3:

Remarks to the Author:

This paper reports the structure of the Zika virus (ZIKV) particle to 3.4Å resolution, which was achieved by performing “sub-volume refinement” from a lower resolution (4Å) icosahedrally averaged cryo-EM map. Although the structure of the ZIKV particle from a different strain has been reported to 3.1 Å resolution, the value of this paper is the identification of two lipids tightly bound between the C-terminal α -helices of the stem of protein E and of protein M. These lipids had not been noticed earlier, although the density for one of them termed “Y shaped” lipid in between the α -helices of the E stem, is present in the 3.1Å resolution structure of the other strain of ZIKV reported earlier. The authors propose that the second lipid, termed “crook shaped” and bound in between the TM helices of E and M, is present only in virions grown in mammalian cells (as is the case here), and not in insect cells, as is the case for the other strain of ZIKV.

The Cryo-EM analysis is of very good quality and the interpretations appear to be sound. Although the nature of the identified lipids was not identified, the identification of the lipid-specific pockets is important, as the residues lining the pockets are conserved across many pathogenic flaviviruses and appear essential for particle morphogenesis and may indeed provide a new handle to develop therapeutic agents against pathogenic flaviviruses. I therefore support publication, provided that the authors address the issues below.

1-The introduction is not clear concerning M and prM. As described the reader understands that they are two different proteins. The presence of prM during assembly, and its cleavage later on into “pr” and “M” should be explained at the start, the first time protein M is mentioned.

2-Line 168: cholesterol does not have an aliphatic chain, which rules out the possibility that the identified “crooked” lipid is cholesterol.

3-Were the E W474 mutants that were lethal (W474A, W474H, W474R) tried in insect cells, which does not incorporate the “crook-shaped” lipid? As the authors show that this lipid is not incorporated when the virus is grown in insect cells, if the mutation is also lethal it might mean that it is just not possible to introduce a polar residue at this position, as this residue is in the trans-membrane region, and not because of the specific lipid pocket.

4- Mention the preprint at doi: <https://doi.org/10.1101/2020.06.07.138669> on the Spondweny virus structure and the lipid pocket described there – also, it is important to compare how the virus was propagated in both studies.

Comments of Reviewer #1, General

In this manuscript, matured Zika virus from Vero-E6 was reconstructed to a 3.4 Å map. Two distinct lipid moieties were identified in a pocket formed by M and E transmembrane helices near the inner leaflet and between two helices of E protein near the outer leaflet, respectively. The density features of Y-shaped lipid can be found but ignored in a previously published map. Using Structure-based mutagenesis, the author showed both lipid moieties play important role in the virus life cycle. The finding is new and interesting.

Major comments

1. The mutagenesis study of E W474 indicates a hydrophobicity of the newly identified inner leaflet pocket. It seems W474 is part of the transmembrane helix, which explains why it requires hydrophobicity. Thus, the hydrophobicity is not just for the crooked lipid. In addition, only virus grown in mammalian cells had the crooked lipid density arising from the inner leaflet also indicates that crooked lipid is not that important since the virus from other cells do not need it. Thus, "Results show that lipids play an essential role in the ZIKV assembly pathway revealing a new target of lipid based antiviral drug development" is not convincing. However, the mutagenesis study of the function of Y-shaped lipid seems more convince to me. Perhaps, Y-shaped lipid plays an essential role in the ZIKV assembly pathway.

We thank the reviewer for this insight. Although it is important to note that the E W474 is part of the transmembrane helix, here we have shown that it is critically positioned at the lipid water interface. Based on our results and the previous literature on interfacial Tryptophan (Trp) residues, we propose that the W 474 present at the viral inner membrane proximal region of the TM helix of E protein has important roles in lipid binding in mammalian cells in particular and virus assembly in general. The aromatic amino acids Trp and Tyrosine (Tyr) are amphipathic due to their polar and non-polar character. In this study we were able to produce infectious virus with non-polar phenyl alanine and polar Tyr at the position of W 474. This observation can be attributed to the property of Trp side chain to form weak hydrogen bonds through the indole nitrogen that is often solvent-exposed. The indole NH hydrogen bond formation is essential for anchoring Trp residues to the membrane-water interface regions (Sun et al 2008). Additionally, Trp has the most significant free energy for partitioning into the head group region of membrane bilayers (Wimley et al 1996). Therefore, the Trp side chains that prefer to reside at the membrane interface are driven by a balance of the hydrophobic effect and interfacial binding to polar lipid moieties. Interactions with interfacial and glycerol backbone polar moieties, particularly with H-bonding with lipid carbonyl and phosphate components, create forces that pull the Trp side chains towards the interface (deJesus et al 2012). Due to these properties, Trp exhibits a strong preference to be at the ends of transmembrane alpha helices near the membrane-water interfacial regions of membrane proteins (Arkin et al 1998).

In addition, several studies have shown that the aromatic Trp residue preferentially residing near the lipid–water interface of the TM helix has a significant anchoring role

besides being part of the TM helix (deJesus 2012). Model peptide studies have shown that the position of Trp at the ends of TM helices can influence the mismatch between the hydrophobic length of the peptide and the bilayer thickness, which can alter the helix tilt angle (de Planque 2003). A Trp residue in the human thrombopoietin receptor at the transmembrane–cytosolic junction has been shown to extend the hydrophobic length and tilt the TM helix by partitioning into the bilayer (Defour 2013). Thus, the interfacial aromatic residues, such as Trp or Tyr situated at the lipid-water interface, are essential for the insertion, stability, and dynamics of transmembrane helices (McKay 2018). Interactions of Trp residues from TM helices at the viral membrane have been observed in other enveloped viruses. A conserved Trp residue, W409 from TM helix of the envelope protein E1 has been shown to be part of a pocket that coordinates an unknown pocket factor near the viral membrane in Sindbis virus (Chen 2018). The pocket factor molecule has been shown to stabilize the pocket by forming several hydrophobic interactions with surrounding proteins, including the amino acids W409 from the E1 TM helix (Chen 2018). Based on these studies, we propose that the Trp 474 residue of ZIKV E protein interacts with the pocket factor through hydrophobic interactions while contributing to additional functions related to virus assembly at the interfacial region. We have clarified our conclusions to differentiate between the critical position of W474 and the incorporation of the crooked and Y-shaped lipids. Similarly, we have modified our conclusion about the necessity of incorporation of the crooked lipid during assembly.

We have introduced clarification in the text for these two points. 1) that the Trp 474 resides at the lipid-water interface, 2) potential function depending on this Trp 474 is stability of the transmembrane helix and dynamics such as assembly. Please see lines 201-213.

2. The authors used a single particle reconstruction approach combining with subvolume refinement to overcome resolution limitations of matured Zika virus and improved the resolution of the map from 4.0 to 3.4 Å. The authors think that the resolution of matured Zika virus is limited by heterogeneity as seen in the reference #10 and #12 in the manuscript. However, in reference 12, the matured Zika virus has been refined to 3.1 Å without subvolume refinement. In addition, a most recent preprint manuscript titled by “A high resolution view of an adolescent flavivirus” on bioRxiv reported matured SPONV and DENV at 2.6 Å and 3.1 Å, respectively. It seems there is no such a limited resolution, since the matured Flaviviruses are rigid enough to achieve high resolution. Thus, it is not suitable to use “resolution limitations” here, probably “improves the resolution” is more suitable.

We thank the reviewer for the helpful comments. We have changed the use of “resolution limitations” to “improves the resolution”, as was suggested. Furthermore, the type of heterogeneity in our particles was not well described or illustrated, and this shortcoming has been addressed. Nearly every particle has one or more asymmetric patches of immature spikes. We also neglected to compare the way our particles were produced so that the reader could understand the differences. In this work the particles were propagated in Vero-E6 cells, which leads to patches of uncleaved prM on the particles. However, when

ZIKV is produced in mosquito or Vero-furin cells, the resulting particles are much more homogeneous (Sirohi 2016). We have now included a clear description of the heterogeneity specific to our preparation of virus which is illustrated with representative micrographs (Fig 1) and with the 2D classification in Extended Data figure 1 (see line 87-92). Along with the better description of the asymmetry and heterogeneity, it is now made clearer why we used special handling of our preparation instead of relying on extended refinement and icosahedral symmetry (please see line 99-105 and in Discussion 160-169).

Incomplete cleavage of PrM resulted in heterogeneity of viruses by leaving some of the PrME immature spikes on the mature viruses. However, such particle can be partially avoided during particle picking or later excluded by a careful 3D classification procedure. In this manuscript, ~34,000 particles are too many to achieve a 4.0 Å 70nm icosahedral virus map, which indicates a lot more virus particles can be excluded from refinement. However, instead, the authors used 3D classification of subvolume to get rid of bad particles. As mentioned in the manuscript, only ~9181 (1%, 9181/30/34000=1%) subvolume was selected for refinement. It seems to me that it is not an efficient way to calculate matured Flaviviruses. During the refinement of the whole virus, the half-matured half-immatured particle is very possibly assigned by a wrong Euler angle. In this case, the local refinement of the subvolume of the matured part of the virus cannot help to find the right Euler angle. Thus, the reason for resolution improvement is different from fixing the flexibility problem of the virus. Therefore, the discussion part “each raft (consisting of three EM dimer pairs) may not maintain strict icosahedral symmetry relative to its neighbor” is possibly incorrect.

We thank the reviewer for their technical feedback and questions. As described by Reviewer #1, 3D classification is used to sort out the best of the icosahedral virus particles, eliminating partially mature/immature and otherwise deformed particles. However, due to the gross overall presence of incomplete cleavage of these mammalian tissue culture derived virus particles, we could not take the traditional approach to sort out good particles from bad particles isolating a homogeneous sub population of mature particles for a high-resolution icosahedral reconstruction. We have also included a better description of the characteristics of mammalian derived virus versus other production methods.

For the ideal dataset, we typically extract subvolumes followed by refinement. This was also not sufficient for the reasons pointed out by our Reviewer #1 as the incorrect Euler angle can be assigned. To overcome this problem after extracting subvolumes, an initial 3D classification was performed to remove subvolumes that were too divergent from expected orientation. Only after this step did we perform local refinement to attain higher resolution. Clarification has been added to Methods.

We agree with the reviewer’s conclusions that the resolution improvement is independent of the flexibility of the virus and have modified the text accordingly.

Furthermore, we have added to our figures, modified our flow chart, and clarified the text to explain our approach and why we took that approach.

Minor comments

1. The Y-shaped lipid was found in SPONV and DENV (bioRxiv preprint doi: <https://doi.org/10.1101/2020.06.07.138669>). The preprint was posted on June 7, 2020. They do not mention the other molecule. It may worth to compare.

Although we like this plan, there is no map in the preprint available to which we can compare. However, we have compared other aspects of this finding with our work as suggested by the Reviewers. Text has been modified line 171-179.

2. “Previously, structural studies to address these unknowns have focused on traditional processing methods, including increased particle number and higher magnification to overcome global and local heterogeneity. However, these approaches will not work when faced with gross heterogeneity. Here, we used a cryo EM reconstruction approach in which subvolumes of the virus were extracted and refined to 3.4 Å resolution.”

We have modified this statement in the text to be clearer about our approach and the reason for it.

Although, the package used here is custom implementation of the localized reconstruction approach, the idea here separating the virus into blocks and locally refining the center and orientation parameters of each block separately using the densities of target protein overlapping with the densities of other proteins (without density subtraction) to overcome the flexibility of the icosahedral viruses is closer to that of “block-based” reconstruction method. In the package of the “block-based” reconstruction method, it provided a tool to put the “subvolume” together to form an intact virus. Both “block-based” and “localized method” are well established to deal with flexibility and heterogeneity. It is better to mention and cite both “block-based” and “localized method” methods in the introduction.

We thank the Reviewer for pointing out this omission on our part and we have cited both methods and the corresponding publications in the resubmission in the introduction, discussion, and methods (Please see Lines 99-105; lines 78-80; lines 283-284).

Comments of Reviewer #2, General:

The manuscript by DiNunno et.al., identifies the presence of lipid moieties that insert and possibly interact with the transmembrane helices (TM) of E and M proteins of Zika virus. The authors have followed up their structural results with point mutations in the TM regions of E and M proteins to identify critical residues that are likely to be important for interaction with the lipid moieties by analyzing the effect of the mutations on viral infectivity through plaque assays. The manuscript is overall written clearly and concisely. Few comments and questions for the authors are below:

Major comments

1. *The authors have used sub-volume classification and reconstruction to improve the resolution of the map beyond what they achieved by icosahedral averaging alone. This helped them assign side-chain densities with more confidence in the structure. However, the explanation of the sub-volume classification procedure in writing and in extended data figure 1 is a bit confusing. In the text, the explanation implies that sub-volumes of 2 asymmetric units related by 2-fold were selected as a unit, mapped back to their locations in original 2D projections and then were used for subsequent 3D classification of the sub-particles individually. But extended figure 1 shows the whole virus being split into 6 classes. Extended data figure 2 is bit more clearer and seems to align with the local sub-volume classification routine. As this procedure is an important point in the manuscript, it needs to be more clearly explained.*

We thank the reviewer for the feedback on the unfortunate confusion caused by Extended Data Figure 1. Extended data figure 1 has been modified to represent accurately the sub-volume classification routine used and the legend expanded for clarity.

The subvolume chosen for extraction corresponds to three M and E dimer pairs and is more clearly described in the text and the figure legends. Extended Data Figure 1 has been changed for added clarity and the classes are now appropriately labeled as raft subvolumes within the flow-chart.

2. *Considering that there is a higher resolution reconstruction of Zika virus at 3.1 angstroms available; the authors have used local reconstruction to push their map resolution to 3.4 angstroms to identify amino acid orientations correctly. It would be interesting if the authors compared the local resolution at the TM regions between the maps. Does their approach improve local resolution at the TM region for better model building? Extended data figure 2, shows local resolution variation in top view but not in side-view which may also help to see how the local resolution fares in the TM regions.*

We have included four alternative views of the local resolution of our map which shows that the TM region at a resolution of about 3.4 Å has not improved enough to change the build, in a new figure (Extended Data Figure 2). We did download from EMDDB and attempt the evaluation of the unfiltered half-maps of the 3.1 Å ZIKV structure using both MonoRes and ResMap. Unfortunately, the manual intervention of ResMap's pre-whitening routine introduces significant variability in local resolution reporting, whereas MonoRes tends to relatively under-report estimates (Luis Vilas, 2018). The variability between different local resolution platforms has been well-evaluated and published (Luis Vilas, 2018). This known variability makes it challenging to conclude the absolute local resolution relative to a map, without introducing too much user bias, particularly within such a small resolution range (3.1-3.4 Angstroms).

3. *The authors have compared their ZIKV reconstruction with other ZIKV*

reconstructions, one at 3.1 angstroms and the other at 3.7 angstroms resolution. They observe that the lipid density arising from the inner leaflet is only seen in viruses grown in mammalian cells and not seen in viruses grown from mosquito cells. Have the authors looked at high resolution structures of other flaviviruses apart from Zika to see if these lipid moieties are seen in other flaviviruses also? The authors have done sequence alignments to show the conservation of important residues in the TM regions of M and E protein among flaviviruses, it would be more informative to see if other maps show similar lipid densities or not.

We did look for the lipid density in the 3.5Å (3J27) dengue virus (Zhang 2012) and 3.9 Å (5O6A) of tick-borne encephalitis virus (Fuzik et al 2018) maps, but in neither case was the density good enough to conclude the presence or absence of bound lipid. This comparison has been included in the Discussion, see line 183-186.

4. Overall, figures need to be annotated more clearly and legibly, so that they are more self-explanatory. Example fig.1 – none of the symmetry axes are labeled nor the different colored layers of the cross-section given a scale bar or notations except in the figure legend. In figure 2, it would be nice to label the central panels that are from different EMDB depositions, so that it is clear what the figure is portraying. Similar issues with extended data figures also which lack clear notations. It will also be helpful if the same coloring scheme for E and M proteins are followed across figures.

We thank the reviewer for their criticism and believe these suggestions will strengthen the clarity and communication of this finding. The figures and figure legends have been reworked.

Comments of Reviewer #3, General:

This paper reports the structure of the Zika virus (ZIKV) particle to 3.4Å resolution, which was achieved by performing “sub-volume refinement” from a lower resolution (4Å) icosahedrally averaged cryo-EM map. Although the structure of the ZIKV particle from a different strain has been reported to 3.1 Å resolution, the value of this paper is the identification of two lipids tightly bound between the C-terminal α -helices of the stem of protein E and of protein M. These lipids had not been noticed earlier, although the density for one of them termed “Y shaped” lipid in between the α -helices of the E stem, is present in the 3.1Å resolution structure of the other strain of ZIKV reported earlier. The authors propose that the second lipid, termed “crook shaped” and bound in between the TM helices of E and M, is present only in virions grown in mammalian cells (as is the case here), and not in insect cells, as is the case for the other strain of ZIKV. The Cryo-EM analysis is of very good quality and the interpretations appear to be sound. Although the nature of the identified lipids was not identified, the identification of the lipid-specific pockets is important, as the residues lining the pockets are conserved across many pathogenic flaviviruses and appear essential for particle morphogenesis and may indeed provide a new handle to develop therapeutic agents against pathogenic flaviviruses. I therefore support publication, provided that the authors address the issues

below.

Major comments:

1-The introduction is not clear concerning M and prM. As described the reader understands that they are two different proteins. The presence of prM during assembly, and its cleavage later on into “pr” and “M” should be explained at the start, the first time protein M is mentioned.

Thank you to the Reviewer for pointing out this problem. The introduction has been modified to clarify this issue. We have added a sentence ‘The M and pr (93 amino acids) proteins are generated from a precursor membrane (prM, 168 amino acids) protein by the host furin protease cleavage at the pr-M junction. Please see lines 37-43.

2-Line 168: cholesterol does not have an aliphatic chain, which rules out the possibility that the identified “crooked” lipid is cholesterol.

We thank the reviewer for making this important observation. We have corrected this in the text in Line 183 where we deleted the following text “and there are likely candidates such as cholesterol.”

3-Were the E W474 mutants that were lethal (W474A, W474H, W474R) tried in insect cells, which does not incorporate the “crook-shaped” lipid? As the authors show that this lipid is not incorporated when the virus is grown in insect cells, if the mutation is also lethal it might mean that it is just not possible to introduce a polar residue at this position, as this residue is in the trans-membrane region, and not because of the specific lipid pocket.

We thank the reviewer for this comment to test the lethal mutations in mosquito cells. It is not possible to transfect our CMV promoter-launched ZIKV cDNA clones into insect cells due to the incompatibility of mammalian expression promoter; however, likely the mutation would be lethal because of the location at the lipid water interface and its likely involvement in assembly.

We did introduce a tyrosine into this position, which was not lethal. This has been made clearer in the text, please see lines 201-213. Specifically the Reviewer’s comment added importance to the discussion.

4- Mention the preprint at doi: <https://doi.org/10.1101/2020.06.07.138669> on the Spondweny virus structure and the lipid pocket described there – also, it is important to compare how the virus was propagated in both studies.

We thank the Reviewer for this suggestion and have included the comparison of lipid binding observed in Spondweni virus in the manuscript. We have also included the propagation. Please see lines 176-183.

References

1. McKay 2018, Control of Transmembrane Helix Dynamics by Interfacial Tryptophan Residues PMID: 29874612
2. deJesus 2012 The role of tryptophan side chains in membrane protein anchoring and hydrophobic mismatch PMID: 22989724
3. Defour 2013 Tryptophan at the transmembrane-cytosolic junction modulates thrombopoietin receptor dimerization and activation PMID: 23359689
4. Arkin IT, Brünger AT (1998) Statistical analysis of predicted transmembrane α -helices. *Biochim Biophys Acta* 1429(1):113–128. PMID: 9920390
5. Wimley WC, White SH (1996) Experimentally determined hydrophobicity scale for proteins at membrane interfaces. *Nat Struct Biol* 3(10):842–848. PMID: 8836100
6. de Planque MRR, Killian JA (2003) Protein-lipid interactions studied with designed transmembrane peptides: Role of hydrophobic matching and interfacial anchoring. *Mol Membr Biol* 20(4):271–284. PMID: 14578043
7. Sun (2008) The preference of tryptophan for membrane interfaces: insights from N-methylation of tryptophans in gramicidin channels PMID: 18550546
8. Chen (2018) Implication for alphavirus host-cell entry and assembly indicated by a 3.5Å resolution cryo-EM structure PMID: PMC6294011
9. Renner (2020) A high resolution view of an adolescent flavivirus. <https://doi.org/10.1101/2020.06.07.138669>
10. Vilas, Jose Luis, Josué Gómez-Blanco, Pablo Conesa, Roberto Melero, José Miguel de la Rosa-Trevín, Joaquin Otón, Jesús Cuenca, et al. “MonoRes: Automatic and Accurate Estimation of Local Resolution for Electron Microscopy Maps.” *Structure (London, England: 1993)* 26, no. 2 (06 2018): 337-344.e4. <https://doi.org/10.1016/j.str.2017.12.018>.

Reviewers' Comments:

Reviewer #1:

Remarks to the Author:

My comments were addressed in the revised manuscript.

Xinzheng Zhang

Reviewer #2:

Remarks to the Author:

The authors have provided a revised manuscript with better detail and highlighting the importance of their results in conjunction with results available from other flaviviruses. I only have a few minor comments:

1. Although the authors have improved notations in the figures, in Fig.2, the different panels are rendered differently. Panel A shows only ribbons on the left but then only density rendering on the right zoom in. The other panels have density and structure fitted into it. A uniform rendering is needed for ease of visualization.
2. In extended data figure 4, the distance values are in yellow making it very hard to read and should be modified.

Comments of Reviewer #1, General

Reviewer #1 (Remarks to the Author):

My comments were addressed in the revised manuscript.

Comments of Reviewer #2, General

Reviewer #2 (Remarks to the Author):

The authors have provided a revised manuscript with better detail and highlighting the importance of their results in conjunction with results available from other flaviviruses. I only have a few minor comments:

1. Although the authors have improved notations in the figures, in Fig.2, the different panels are rendered differently. Panel A shows only ribbons on the left but then only density rendering on the right zoom in. The other panels have density and structure fitted into it. A uniform rendering is needed for ease of visualization.

The Panels have been changed to ribbon for uniformity.

2. In extended data figure 4, the distance values are in yellow making it very hard to read and should be modified.

The font has been outlined in black to make them more visible.